# The Neuroprotective Potential of Seed Extract from the Indian Trumpet Tree Against Amyloid Beta-Induced Toxicity in SH-SY5Y Cells

**DOI:** 10.3390/ijms26136288

**Published:** 2025-06-29

**Authors:** Nut Palachai, Benjaporn Buranrat, Parinya Noisa, Nootchanat Mairuae

**Affiliations:** 1Biomedical Research Unit, Faculty of Medicine, Mahasarakham University, MahaSarakham 44000, Thailand; nut.p@msu.ac.th (N.P.); buranrat@gmail.com (B.B.); 2School of Biotechnology, Institute of Agricultural Technology, Suranaree University of Technology, Nakhon Ratchasima 30000, Thailand; p.noisa@sut.ac.th

**Keywords:** Alzheimer’s disease, Indian trumpet tree, *Oroxylum indicum* (L.) seed, neuroprotection, oxidative stress, plant extraction, protein kinase B, cAMP response element-binding protein, mitogen-activated protein kinases

## Abstract

Alzheimer’s disease (AD) is a progressive neurodegenerative disorder with an unclear etiology. Multiple factors, including oxidative stress and the accumulation of amyloid beta (Aβ) protein in the brain, contribute to neuronal damage. This study investigated Aβ-induced oxidative stress and cellular damage in SH-SY5Y cells, as well as the neuroprotective potential of Indian trumpet tree seed extract (ITS). SH-SY5Y cells were co-treated with Aβ_(25–35)_ (20 µM) and ITS extract at concentrations of 25 and 50 µg/mL. Cell viability, reactive oxygen species (ROS), malondialdehyde (MDA) levels, and the enzymatic activities of catalase (CAT), superoxide dismutase (SOD), and glutathione peroxidase (GSH-Px) were assessed. The expression levels of B-cell lymphoma 2 (Bcl-2) and caspase-3, along with the phosphorylation levels of protein kinase B (Akt), extracellular signal-regulated kinases 1 and 2 (ERK1/2), and cAMP response element-binding protein (CREB), were also evaluated. ITS extract at concentrations of 25 and 50 µg/mL significantly improved SH-SY5Y cell viability following Aβ-induced damage; reduced ROS and MDA levels; and enhanced CAT, SOD, and GSH-Px activities. In addition to upregulating Bcl-2 expression, ITS downregulated caspase-3 expression and increased the phosphorylation of Akt, ERK1/2, and CREB. High-performance liquid chromatography (HPLC) analysis identified baicalin, baicalein, and chrysin as major phenolic compounds in ITS extract. In conclusion, ITS extract attenuated Aβ-induced oxidative stress, enhanced antioxidant defenses and cell viability, suppressed apoptotic signaling, and activated key neuroprotective pathways. These findings provide new insights into the neuroprotective potential of ITS extract; however, further in vivo studies are needed to validate its clinical applicability.

## 1. Introduction

Alzheimer’s disease (AD) is a progressive neurological ailment that affects the elderly and causes cognitive impairment, memory loss, and daily activity problems [1]. AD accounts for 60–80% of dementia cases worldwide and is the sixth largest cause of death among 65-year-olds [2]. Around 6.7 million Americans have AD, and worldwide instances are over 55 million and expected to quadruple by 2050 [3]. This rising prevalence and an aging global population make AD a major challenge for healthcare systems [4]. The underlying pathophysiology of AD is complex and involves multiple factors that are not yet fully understood [1]. The histological features of AD include amyloid beta (Aβ) plaques and intraneuronal neurofibrillary tangles composed of hyperphosphorylated tau protein [5,6,7]. These pathological changes disrupt neuronal communication and ultimately lead to neuronal death [7]. Oxidative stress is another hallmark of AD pathology [6,8], characterized by the excessive production of reactive oxygen species (ROS) in both AD patient brains and transgenic mouse models [9]. There is a substantial body of research that has proven the presence of widespread oxidative damage in AD brains, which is intimately linked to the buildup of Aβ plaques and neurofibrillary tangle-like structures [10,11]. Furthermore, it is worth noting that oxidative stress, in conjunction with key pathogenic aspects of AD, such as tau hyperphosphorylation, Aβ aggregation, and synaptic dysfunction, plays a role in contributing to a self-reinforcing cycle of neuronal death [10,11,12]. 

Currently approved treatments for AD by the United States Food and Drug Administration (FDA) include cholinesterase (AChE) inhibitors and N-methyl-D-aspartate (NMDA) receptor antagonists [13], which provide only symptomatic relief and do not halt disease progression. This limitation underscores the need for novel therapeutic approaches that target disease-modifying pathways. Given the central role of oxidative stress in AD progression, antioxidant therapies have been investigated as potential treatment strategies. These therapies aim to reduce ROS levels and protect mitochondrial function, thereby preventing or slowing neuronal damage. The current study underscores the promise of herbal medicine as a viable therapeutic method due to its richness in bioactive constituents, including polyphenols, flavonoids, tannins, and glycosides, which augment its therapeutic efficacy across multiple diseases [14,15]. Recent research indicates that these herbal plants confer neuroprotective effects through many mechanisms, including antioxidant, anti-apoptotic, anti-amyloidogenic, anti-tau, and anti-inflammatory characteristics, as well as free radical scavenging [14,15,16]. These characteristics collectively augment their efficacy in the prevention and treatment of AD.

The Indian trumpet tree (*Oroxylum indicum* (L.) Kurz), widely distributed across Southeast and South Asia, is named for the distinctive shape of its flowers [17]. This edible plant has been used in traditional Thai medicine for centuries, with all parts of the plant—including the stem bark, pods, leaves, and seeds—demonstrating various pharmacological activities, including antioxidant [18,19], anti-inflammatory [20,21,22], and neuroprotective effects [23,24,25]. Key phytoconstituents in the Indian trumpet tree include baicalein, baicalin, chrysin, and oroxylin A [21,24,26,27]. Notably, Indian trumpet tree seed (ITS) extract contains the highest baicalin and baicalein content [26]. Baicalin (baicalein-7-O-glucuronide) and its aglycone, baicalein (5,6,7-trihydroxyflavone), exhibit some of the most potent biological activities among flavonoids, particularly in relation to neurodegenerative diseases such as depression and cognitive impairment [28,29,30]. Prior studies indicate that baicalin diminishes cerebral inflammation, oxidative stress, and apoptosis while concurrently enhancing neurogenesis and ameliorating mitochondrial dysfunction [29]. Moreover, baicalein and baicalin have been documented for their inhibitory activity on AChE enzyme, anti-amyloid protein aggregation effects, and the attenuation of Aβ_(1–42)_-induced apoptosis in SH-SY5Y cells [31,32,33,34].

Despite the abundance of baicalin and baicalein in ITS, there is no scientific evidence examining the neuroprotective properties of ITS extract in an AD model yet. Therefore, the present study aimed to investigate the neuroprotective potential of ITS extract against oxidative stress and cellular damage induced by Aβ_(25–35)_ in SH-SY5Y cells and to elucidate the underlying mechanisms of its protective effects. The SH-SY5Y human neuroblastoma cell line was chosen as a model of this study because it is a well-established in vitro model for researching neurotoxicity and neuroprotective pathways, particularly in AD research [35]. An Aβ_(25–35)_ fragment was selected to induce oxidative stress and neurotoxicity because it is a synthetic fragment of the full-length Aβ peptide that preserves key neurotoxic characteristics of Aβ_(1–42)_, including the ability to trigger oxidative stress, mitochondrial dysfunction, and apoptosis [36]. 

## 2. Results

### 2.1. The Impact of Amyloid Beta Aβ25-35 on the Survival of SH-SY5Y Cells

To evaluate the cytotoxic effects of Aβ_(25–35)_ on SH-SY5Y neuroblastoma cells, various concentrations (0–40 µM) were tested. A dose-dependent reduction in cell viability was observed following exposure to Aβ_(25–35)_, as shown in Figure 1A. At a concentration of 20 µM, a significant decrease in cell survival was detected (*p* < 0.01 compared with the control group), indicating that Aβ_(25–35)_ induces substantial neurotoxicity at this level. Based on this finding, 20 µM was selected as the optimal concentration for inducing neuronal damage and was used in all subsequent experiments to evaluate the neuroprotective efficacy of the test compounds.

### 2.2. The Effects of Indian Trumpet Tree Seed (ITS) on the Viability of SH-SY5Y Cells

To evaluate the effects of ITS extract on SH-SY5Y cell viability, the cells were treated with various concentrations of the extract, ranging from 0 to 100 µg/mL. The results indicated that ITS extract did not significantly affect cell viability at concentrations of 12.5, 25, and 50 µg/mL compared with the control group. However, 100 µg/mL was found to be significantly (*p* < 0.01) toxic to SH-SY5Y cells, as shown in Figure 1B. Therefore, the maximum non-toxic concentrations of ITS extract (25 and 50 µg/mL) were selected for use in subsequent experiments.

### 2.3. ITS Protects Against Aβ_(25–35)_-Induced Cytotoxicity in SH-SY5Y Cells

The protective effect of ITS against Aβ_(25–35)_-induced cytotoxicity was assessed by treating SH-SY5Y cells with 20 µM Aβ_(25–35)_ in the presence or absence of 25 and 50 µg/mL ITS extract for 24 h. As shown in Figure 2A, exposure to Aβ_(25–35)_ resulted in a significant reduction in cell density along with distinct morphological changes compared with the untreated control group. These alterations included cellular shrinkage and loss of neurite projections, indicating cytotoxic effects. Moreover, the MTT assay results depicted in Figure 2B demonstrate that Aβ_(25–35)_ treatment significantly reduced cell viability (*p* < 0.01), thereby confirming its pronounced neurotoxic effect. By contrast, co-treatment with ITS extract at 25 and 50 µg/mL significantly attenuated Aβ-induced cytotoxicity. Cells treated with ITS showed improved morphology and increased cell density (Figure 2A), suggesting protective effects against Aβ toxicity. Quantitative analysis further revealed that ITS extract significantly enhanced cell viability in a dose-dependent manner (*p* < 0.01), highlighting its strong cytoprotective potential (Figure 2B).

### 2.4. ITS Decreases Aβ_(25–35)_-Induced ROS Generation in SH-SY5Y Cells

It has been suggested that the toxicity of Aβ_(25–35)_ might be ascribed to the production of an excessive amount of ROS, which ultimately results in oxidative stress and damage to cells. This study evaluated the effect of ITS extract on intracellular ROS generation induced by Aβ_(25–35)_. As shown in Figure 3, exposure to Aβ_(25–35)_ for 24 h significantly increased ROS levels compared with the untreated control group (*p* < 0.01), confirming its role in promoting oxidative stress. 

However, co-treatment with ITS extract at concentrations of 25 and 50 µg/mL significantly reduced ROS accumulation in a concentration-dependent manner (*p* < 0.01). These findings suggest that ITS extract exerts antioxidant effects, contributing to its protective role against Aβ-induced oxidative stress.

### 2.5. Effects of ITS on Oxidative Stress Status

Table 1 presents the effects of ITS extract on oxidative stress markers. SH-SY5Y cells treated with Aβ_(25–35)_ showed a significant increase in MDA levels compared with the control group (*p* < 0.05), indicating enhanced lipid peroxidation. This was accompanied by a significant reduction in the activities of SOD and GSH-Px (*p* < 0.05), while CAT activity remained unchanged in Aβ_(25–35)_-treated cells.

By contrast, co-treatment with Aβ_(25–35)_ and ITS extract significantly reduced MDA levels and markedly increased the activities of CAT, SOD, and GSH-Px compared with the Aβ_(25–35)_ group. These findings suggest that ITS extract mitigates oxidative stress by restoring antioxidant enzyme activities and reducing lipid peroxidation.

### 2.6. ITS Reduces Caspase-3 Expression Induced by Aβ_(25–35)_

Caspase-3 is a crucial executioner enzyme in the apoptotic pathway and is widely recognized as a key biomarker of neuronal apoptosis [37]. In this study, the effect of ITS extract on caspase-3 expression was examined to better elucidate its neuroprotective mechanism. As illustrated in Figure 4, treatment with Aβ_(25–35)_ for 24 h significantly increased caspase-3 expression compared with the untreated control group (*p* < 0.05). However, co-treatment with ITS extract at both 25 and 50 μg/mL significantly suppressed caspase-3 expression in a concentration-dependent manner relative to Aβ_(25–35)_ treatment alone (*p* < 0.01). These findings suggest that ITS extract attenuates Aβ_(25–35)_-induced neuronal apoptosis by inhibiting caspase-3 activation, further supporting its potential as a neuroprotective agent.

### 2.7. ITS Increases ERK1/2 and Akt Phosphorylation

Neuronal development, plasticity, and survival are regulated by the ERK/MAPK and Akt signaling pathways [27,38]. The present study investigated whether ITS extract modulates these pathways to mitigate the neurotoxicity induced by Aβ_(25–35)_. The results demonstrated that treatment with Aβ_(25–35)_ significantly reduced the phosphorylation levels of Akt (p-Akt; Figure 5A) (*p* < 0.01) and ERK1/2 (*p*-ERK1/2; Figure 5B) (*p* < 0.05) compared with the control group, indicating impaired signaling activity crucial for neuronal function.

By contrast, co-treatment with ITS extract significantly restored the phosphorylation levels of both Akt and ERK1/2 in a concentration-dependent manner relative to Aβ_(25–35)_ treatment alone (*p* < 0.01).

### 2.8. ITS Enhances CREB Phosphorylation

The present study examined the ability of ITS extract to modulate CREB phosphorylation, a downstream target of Akt and ERK1/2 [39], in SH-SY5Y cells exposed to Aβ_(25–35)_. The results demonstrated that treatment with Aβ_(25–35)_ reduced CREB phosphorylation compared with the control group (*p* < 0.05). However, co-treatment with ITS extract markedly increased CREB phosphorylation levels in a concentration-dependent manner compared with the Aβ_(25–35)_-treated group (*p* < 0.05 and *p* < 0.01) (Figure 6A). These findings suggest that ITS extract not only counteracts the inhibitory effects of Aβ_(25–35)_ on CREB activation but also enhances CREB phosphorylation beyond baseline levels.

### 2.9. ITS Increases Bcl-2 Expression

The transcription factor CREB is a key regulator of the anti-apoptotic gene Bcl-2, which plays a pivotal role in promoting cell survival and inhibiting apoptosis in neurons [40]. The present study demonstrated that exposure to Aβ_(25–35)_ for 24 h significantly decreased Bcl-2 expression (*p* < 0.05) compared with the control group, highlighting the pro-apoptotic effects of Aβ_(25–35)_. Treatment with ITS extract significantly elevated Bcl-2 expression levels in a concentration-dependent manner relative to the Aβ_(25–35)_-treated group (*p* < 0.05 and *p* < 0.01) (Figure 6B). These results indicate that ITS extract alleviates the neurotoxic effects of Aβ_(25–35)_ by augmenting the production of this essential anti-apoptotic protein.

### 2.10. Analysis of Flavonoid Content in the ITS Extract by High-Performance Liquid Chromatography (HPLC)

The HPLC analysis of the ITS extract provided a detailed chromatographic fingerprint, highlighting the presence of distinct bioactive compounds. As shown in Figure 7, the chromatogram revealed four prominent peaks corresponding to oroxin B, baicalin, baicalein, and chrysin, with retention times of 11.16, 18.60, 40.77, and 46.92 min, respectively. These compounds were positively identified by comparing their retention times and spectral characteristics with those of the standard mixture chromatogram shown in Figure 8, ensuring reliable identification. An optimized and validated HPLC method was employed to quantify these compounds in the ITS extract. The analysis determined the concentrations of oroxin B, baicalin, baicalein, and chrysin to be approximately 2.63, 11.33, 8.19, and 3.52 µg/g of extract, respectively.

## 3. Discussion

The present study aimed to determine whether ITS extract protects SH-SY5Y neuronal cells from Aβ-induced oxidative stress and the underlying neuroprotective mechanisms. The result revealed that Aβ_(25–35)_ exhibited concentration-dependent cytotoxicity towards SH-SY5Y cells, which is in line with other results [23,24,41,42]. ITS treatment markedly improved the survival of SH-SY5Y cells exposed to Aβ_(25–35)_, indicating that ITS extract could protect SH-SY5Y cells from damage caused by Aβ. 

According to reports, the development of amyloid plaque in the brains of patients with AD results in increased oxidative stress and ROS generation, which are lethal [9,43,44]. According to our research, cells treated with Aβ_(25–35)_ generated more ROS, which is consistent with other findings [23,24,45]. Following treatment, ITS was found to have an antioxidant effect that, in a concentration-dependent manner, reduced the formation of ROS caused by Aβ. These findings suggested that ITS extract possessed antioxidant properties capable of reducing oxidative damage caused by Aβ_(25–35)_ and highlighted its potential role in alleviating neurotoxicity associated with oxidative stress. Under typical physiological circumstances, the body’s natural antioxidant mechanisms, which include the coordinated action of CAT and SOD, balance the formation of ROS. The first line of defense against free radicals is SOD, which breaks down the superoxide anion radical into oxygen and hydrogen peroxide (H_2_O_2_) [46]. The subsequent detoxification of H_2_O_2_ into water and oxygen is facilitated by enzymes such as CAT and GSH-Px [47]. The present study also explored the impact of ITS on oxidative stress markers, with a particular focus on MDA levels and the activities of key antioxidant enzymes, including CAT, SOD, and GSH-Px. The exposure of SH-SY5Y cells to Aβ_(25–35)_ resulted in a pronounced increase in MDA levels, indicating elevated lipid peroxidation and oxidative stress. This oxidative imbalance was further evidenced by a significant reduction in the activities of SOD and GSH-Px, highlighting a compromised antioxidant defense system. The activity of CAT remained unaffected by Aβ_(25–35)_ treatment, suggesting that the oxidative stress induced by Aβ_(25–35)_ primarily impaired other components of the antioxidant network. By contrast, co-treatment with Aβ_(25–35)_ and ITS extract led to a significant reduction in MDA levels, indicating a mitigation of lipid peroxidation and oxidative damage. Furthermore, the co-treatment resulted in a marked increase in the activities of CAT, SOD, and GSH-Px compared with the Aβ_(25–35)_-treated group, reflecting a substantial enhancement in the cellular antioxidant defense mechanisms. These results underscore the ability of ITS extract to counteract Aβ_(25–35)_-induced oxidative stress by restoring the balance between oxidative damage and the antioxidative defense system. Our findings align with several studies on medicinal plants and their active compounds, which have reported comparable antioxidant and neuroprotective effects in models of Aβ-induced neurotoxicity and other related experimental models. For instance, Huang-Lian-Jie-Du-Tang, a traditional Chinese herbal preparation, has been shown to reduce oxidative stress and increase antioxidant enzyme activities in the hippocampus of Aβ_(25–35)_-treated rats, thereby alleviating oxidative stress and improving learning and memory performance [48]. Similarly, *Lycium barbarum*, a traditional medicinal herb, has been reported to increase antioxidant activities and reduce MDA levels in diabetic rats [49]. Moreover, flavonoid-rich extracts such as those from quercetin have demonstrated the ability to enhance antioxidant defense systems (CAT, SOD, and GSH-Px) and suppress lipid peroxidation in diabetic encephalopathy models, indicating broader applicability in neurodegenerative conditions linked to oxidative imbalance [50]. Many natural antioxidants are derived from plants, particularly phenolics and polyphenolics. These compounds, characterized by hydroxyl groups on their aromatic rings, exhibit strong antioxidant activity by scavenging free radicals, donating hydrogen atoms or electrons, and chelating metal ions [51]. As the most common group of polyphenolic compounds, flavonoids demonstrate a broad spectrum of antioxidant activities against free radical-mediated cellular signaling [51]. Their neuroprotective effects are partly attributed to these antioxidant properties, which help prevent free radical formation by modulating signaling pathways involved in the expression of antioxidant proteins, glutathione synthesis, and the regulation of cell proliferation and survival [51]. These consistent observations across studies lend further support to the therapeutic strategy of using plant-derived flavonoids or polyphenols to counteract oxidative stress-induced neuronal injury in AD. Therefore, our study adds to the growing body of evidence that medicinal plants rich in flavonoids or polyphenols, such as ITS extract, may serve as promising candidates in the development of antioxidative therapies targeting early pathophysiological processes in AD.

Aβ accumulation leads to the initiation of apoptotic signaling cascades, during which caspase-3 serves a pivotal role as an executioner caspase [52]. The impact of ITS on this effector caspase was investigated in the present study. The results indicated that Aβ_(25–35)_ treatment increased caspase-3 levels significantly, and the administration of ITS substantially prevented this increase. One possible explanation for the decrease in caspase-3 levels could be the reduction in ROS generation brought on by ITS therapy. Based on these findings, it can be inferred that ITS extract could reduce the occurrence of neuronal apoptosis caused by Aβ_(25–35)_ by reducing the activation of caspase-3. This further substantiates its potential as a therapeutic agent for neuroprotection.

The ERK/MAPK and Akt signaling pathways serve crucial roles in regulating neuronal differentiation, plasticity, and survival [27,38,53]. The present study explored whether ITS extract modulates these pathways to counteract Aβ_(25–35)_-induced neurotoxicity. The present study revealed that Aβ_(25–35)_ therapy markedly reduced p-Akt and p-ERK1/2 levels, indicating the disruption of vital neural signaling processes. These experimental results are consistent with previous studies demonstrating that Aβ treatment reduces p-Akt and p-ERK levels [25,54,55]. Co-treatment with ITS extracts significantly increased Akt and ERK1/2 phosphorylation levels, indicating that ITS extract could activate the ERK/MAPK and Akt pathways, potentially reversing the neurotoxic effects of Aβ_(25–35)_. This suggests that ITS extract may be used for the treatment of neurodegenerative diseases caused by intracellular signaling system disruption. The activation of Akt and ERK1/2 in neurons has also been linked at the transcriptional level to the phosphorylation of CREB [39], a transcription factor linked to pro-survival through the upregulation of genes such as BDNF and Bcl-2. The present study revealed that Aβ_(25–35)_ therapy markedly decreased CREB phosphorylation, indicating its negative impact on neuronal signaling. These experimental results align with previous studies showing that Aβ treatment decreases p-CREB levels [25,56]. However, co-treatment with ITS extract significantly boosted CREB phosphorylation levels. These results indicated that ITS extract counteracted the inhibitory effects of Aβ_(25–35)_ on CREB activation and elevated CREB phosphorylation levels. ITS extract may promote neuronal survival, synaptic resilience, and neuroprotection by activating CREB-dependent transcriptional processes, offering a viable therapeutic approach.

CREB also regulates Bcl-2, which promotes neuronal survival and prevents apoptosis [40]. In the present study, Aβ_(25–35)_ exposure markedly decreased Bcl-2 expression, indicating pro-apoptotic effects. This experimental result agrees with prior research showing that Aβ treatment decreases Bcl-2 expression [57]. ITS extract treatment resulted in a concentration-dependent increase in Bcl-2 expression compared with Aβ_(25–35)_ treatment. Increased Bcl-2 expression following ITS extract treatment may reduce the neurotoxic effects of Aβ_(25–35)_ by promoting its expression. ITS extract may aid neuronal survival and resilience in pathological settings by balancing pro- and anti-apoptotic signals. These data suggest that ITS extract may be a promising treatment for Aβ-induced neuronal damage and neuroprotection in neurodegenerative disorders.

The present HPLC approach identified baicalin, baicalein, and chrysin as the main active flavonoids in ITS extract, similar to a prior study [21,26]. In vitro and in vivo studies have demonstrated the neuroprotective properties of baicalein [28,29,30,40,41,42]. Baicalein also reduces oxidative stress marker levels, alleviating cognitive impairment caused by chronic cerebral hypoperfusion [57]. In AD and Parkinson’s disease (PD), it reduces oxidative stress, decreases amyloid protein aggregation, suppresses excitotoxicity, stimulates neurogenesis, and is anti-apoptotic and anti-inflammatory [57,58,59]. Specifically, baicalein also significantly reduced ROS generation and alleviated Aβ_(25–35)_-induced cell cycle arrest [59]. It also suppressed neuronal apoptosis by mitigating mitochondrial dysfunction—evidenced by improved membrane potential, reduced Ca^2+^ accumulation, and a lowered Bax/Bcl-2 ratio [59]. Flavonoids such as baicalin are also important. This is mostly derived from the roots of *Scutellaria baicalensis* Georgi, a Chinese medicinal herb [60]. Recent experiments have demonstrated that baicalin protected neurons in both in vitro and in vivo models of neuronal injury [57,58]. Baicalin mitigates neurodegenerative disorders by antioxidative stress, anti-excitotoxicity, anti-apoptotic mechanisms, anti-inflammatory actions, neurogenesis, and the expression of neural protective factors, among other pharmacological activities [57,58]. Chrysin, a bioactive herbal molecule, exhibits a wide range of pharmacological effects, including antioxidant, anti-inflammatory, and neuroprotective properties [61]. Increasing evidence has highlighted the significant role of chrysin in various neurological disorders, such as AD and PD [61]. Chrysin has been shown to exert neuroprotective effects through multiple mechanisms of action, including its antioxidant, anti-inflammatory, and anti-apoptotic functions [61]. In addition, chrysin can indirectly reduce oxidative stress within cells by enhancing the expression levels of key antioxidant enzymes, including SOD, CAT, and GSH-Px [61]. 

From the HPLC assay, our findings highlight the chemical composition of the ITS extract—particularly baicalein and baicalin, which are the most abundant flavonoids in the extract—as key contributors to its antioxidant and neuroprotective properties. The quantification of these markers further supports the use of ITS extract as a therapeutic agent, providing a scientific basis for its efficacy in mitigating Aβ_(25–35)_-induced neurotoxicity. However, several minor peaks were observed in the HPLC chromatogram of the ITS extract, which may represent other potentially bioactive compounds and warrant further investigation. 

The limitation of this study is that it was conducted in vitro using the SH-SY5Y cell model. While this model is widely used for studying neurodegenerative mechanisms, it does not fully replicate the complexity of neuronal networks or the in vivo environment. Therefore, further studies using animal models are essential to validate the neuroprotective potential of ITS extract in a more biologically relevant context. Moreover, future research should focus on isolating baicalin and baicalein from ITS extract to specifically investigate their individual neuroprotective effects.

## 4. Materials and Methods

### 4.1. Chemicals, Reagents, and Antibodies

All the cell culture reagents, including penicillin/streptomycin, were sourced from Hyclone (Logan, UT, USA). The ROS detection kit (Cat. No. 4091-99-0), Aβ_(25–35)_ peptide (Cat. No. A4559-1MG), and MTT assay kit (Cat. No. M5655-1G) were acquired from Sigma-Aldrich (St. Louis, MO, USA). Antibodies targeting total Akt (Cat. No. A17909), phospho-Akt (p-Akt; Cat. No. AP0637), total ERK (Cat. No. A16686), phospho-ERK (p-ERK; Cat. No. AP0974), and actin (Cat. No. AC004) were obtained from Abclonal Biotech Co., Ltd. (Wuhan, China). Antibodies targeting phospho-CREB (p-CREB; Cat. No. AF3189) and BCL-2 (Cat. No. AF6139) were procured from Affinity Bioscience (Nanjing, China). Secondary antibodies, namely anti-rabbit (Cat. No. 34160) and anti-mouse (Cat. No. 31430), along with enhanced chemiluminescence (ECL) detection kits (Cat. No. 34095), were acquired from Thermo Fisher Scientific (Waltham, MA, USA). Supplementary reagents from Sigma-Aldrich comprised RIPA buffer (Cat. No. R0278-50ML), protease inhibitor cocktail (Cat. No. P2714-BTL), and phosphatase inhibitor cocktail (Cat. No. P0044). The bicinchoninic acid (BCA) protein assay kit (Catalog No. 23227) was procured from Thermo Fisher Scientific (Rockford, IL, USA). The malondialdehyde (MDA) detection kit (Catalog No. MAK568) was provided by Merck KGaA (Darmstadt, Germany). The catalase activity assay kit (Cat. No. K773-100) was obtained from BioVision Inc. (Milpitas, CA, USA), and the SOD activity assay kit (Cat. No. 50-190-3738) was procured from Dojindo Molecular Technologies, Inc. (Kumamoto, Japan). The GSH-Px assay kit (Catalog No. E-BC-K096) was procured from Elabscience Biotechnology Inc. (Wuhan, China). 

### 4.2. Plant Material and Extraction

The Applied Thai Traditional Medicine Department, Faculty of Medicine, Mahasarakham University, identified ITS samples obtained from Maha Sarakham Province, Thailand. The Mahasarakham University Faculty of Science Herbarium now possesses a voucher specimen (MSUT 7226). The seeds were dehydrated, weighed, and chopped into smaller pieces to prepare the ethanolic extract. A 7-day maceration in 95% ethanol was subsequently performed on the sample at ambient temperature. The final extract was obtained by filtering the mixture, concentrating it using a rotary evaporator, and finally lyophilizing it [24].

### 4.3. Cell Culture

The SH-SY5Y cell line, derived from a neuroblastoma bone marrow biopsy and displaying neuronal characteristics, was acquired from the American Type Culture Collection (Cat. No. CRL-2266). The SH-SY5Y cells were cultured according to established techniques. The cells were cultured in DMEM with 10% FBS, 1% penicillin–streptomycin, and 1% non-essential amino acids at 37 °C in a humidified environment with 5% CO_2_. For the experiments, cells were inoculated at suitable densities, and the culture media were substituted with fresh medium containing Aβ, with or without ITS treatment [62,63].

### 4.4. Cell Viability Assay

The MTT assay was employed to evaluate the cytotoxic effects of Aβ_(25–35)_ and ITS extract in vitro. SH-SY5Y cells were plated in 96-well plates at a density of 1 × 10^4^ cells per well and cultured under standard conditions. Cells were subjected to treatment with Aβ_(25–35)_ (20–40 μM) and ITS extract (0–100 μg/mL) in serum-free media for 24 h. To evaluate the neuroprotective effects of ITS extract against Aβ_(25–35)_-induced toxicity, cells were treated with Aβ_(25–35)_ alone or in combination with the extract under the same conditions. The media were removed following 24 h of treatment, and the cells were subsequently treated with MTT reagent (0.5 mg/mL) at 37 °C in a 5% CO_2_ atmosphere for 1 h. Following incubation, the medium was discarded, and the purple formazan crystals were solubilized in 100 μL of DMSO. The absorbance was measured at 570 nm with a Synergy-4 plate reader (BioTek; Instruments, Winooski, VT, USA), and the results are presented as a percentage of the control [62,63].

### 4.5. Intracellular ROS Assay

SH-SY5Y cells were seeded into 96-well plates and cultured under standard conditions. After 24 h, the cells were incubated with 10 μM CM-H_2_DCFDA at 37 °C for 30 min in a CO_2_ incubator. Following the incubation, the cells were washed with PBS and treated with Aβ_(25–35)_, either alone or in combination with ITS extract, in serum-free medium for 24 h. The fluorescence intensity of dichlorofluorescein, which reflects ROS levels, was measured using a Synergy HT Multi-Mode Microplate Reader (BioTek; Agilent Technologies, Inc., Winooski, VT, USA), with excitation and emission wavelengths set at 488 and 520 nm, respectively [62].

### 4.6. Determination of Oxidative Stress Status

After the cells were homogenized in 0.1 M potassium phosphate-buffered solution with a pH of 7.4, the changes in the oxidative stress indicators could be examined. Diluting 10 mg of the substance in 50 µL of PBS was part of the homogenization procedure. To determine the protein concentration in the cell homogenates, the Bradford test was employed. Using commercial kits, MDA levels, as well as SOD, CAT, and GSH-Px activity levels, were measured according to the manufacturer’s recommendations.

### 4.7. Western Blotting

SH-SY5Y cells were cultured in 6-well plates at a density of 1 × 10^5^ cells/well. After treatment, proteins were isolated from SH-SY5Y cells using RIPA buffer. The mixture was centrifuged at 10,000× *g* at 4 °C for 10 min, and the supernatant was collected. Protein concentrations were measured with a BCA kit. For separation, 20 µg of protein lysate was mixed with loading buffer, denatured at 95 °C for 5 min, and subjected to SDS-PAGE. Proteins were then transferred to a PVDF membrane, which was blocked with 5% skimmed milk in TBS with 0.1% Tween 20 and incubated overnight at 4 °C with primary antibodies diluted at a 1:1000 ratio. Antibodies against caspase-3, Bcl-2, p-Akt, total Akt, p-ERK1/2, total ERK1/2, and p-CREB were utilized. The antibody against actin was diluted at a 1:5000 ratio. The membranes were rinsed with Tris-buffered saline containing 0.1% Tween 20 and then incubated with secondary antibodies conjugated with HRP for 1 h at room temperature. The protein bands were identified with an enhanced chemiluminescence detection kit, and the results are expressed as fold changes compared with the untreated controls [63].

### 4.8. Analysis of Flavonoid Contents in Crude Extracts Using HPLC

The HPLC method was employed for the quantitative analysis of flavonoids in the crude extracts. A high-performance liquid chromatographic system was utilized, specifically an Accela UHPLC System (Thermo Fisher Scientific, Inc.), which featured a diode array detector surveyor and a column heater. A Pickering C18 column (150 mm × 4.6 mm; 5 μm particle size) was utilized, and this was obtained from Pickering Laboratories. Gradient elution utilized acetonitrile (solvent A), methanol (solvent B), and 0.01% phosphoric acid in water (solvent C), maintained at a constant flow rate of 800 mL/min. The gradient conditions applied were as follows: At 0 min, the concentrations were A = 15%, B = 10%, and C = 75%; at 30 min, A = 19.5%, B = 11.5%, and C = 69%; at 35 min, A = 21%, B = 12%, and C = 67%; at 40 min, A = 31%, B = 15%, and C = 54%; at 60 min, A = 43%, B = 17%, and C = 40%; at 60.1 min, A = 15%, B = 10%, and C = 75%; and at 63 min, A = 15%, B = 10%, and C = 75%. The column temperature was maintained at 25 °C with an injection volume of 10 μL, and detection occurred at 280 nm. The peak area for each standard compound was quantified [64].

### 4.9. Statistical Analysis

Data are presented as the mean ± SEM of at least three independent experiments performed in triplicate. Statistical analysis was conducted using one-way ANOVA testing followed by Bonferroni post hoc tests. *p* < 0.05 was considered to indicate a statistically significant difference. All statistical analyses were performed using SPSS software (version 21.0, IBM Corp., Armonk, NY, USA).

## 5. Conclusions

The present study revealed that ITS extract exhibited protective effects against damage caused by Aβ in SH-SY5Y cells. These protective effects were exerted via different mechanisms. A few examples of these are the reduction in intracellular ROS and MDA levels, the enhancement in antioxidant enzyme activities, the modulation of the caspase-3 pathway, and the activation of essential signaling pathways, including the ERK1/2 and Akt/CREB/Bcl-2 signaling pathways. These findings, when taken as a whole, shed light on the potential of ITS as a neuroprotective agent that employs a comprehensive approach to mitigate the neuronal damage caused by Aβ (as shown in Figure 9). Future research should examine the effects of ITS on in vivo models of AD in order to further evaluate its therapeutic significance. Furthermore, separating active ingredients like baicalin and baicalein for individual analysis may assist in elucidating their distinct roles. These efforts would support the development of ITS-based interventions as potential candidates for AD treatment.

## Figures and Tables

**Figure 1 ijms-26-06288-f001:**
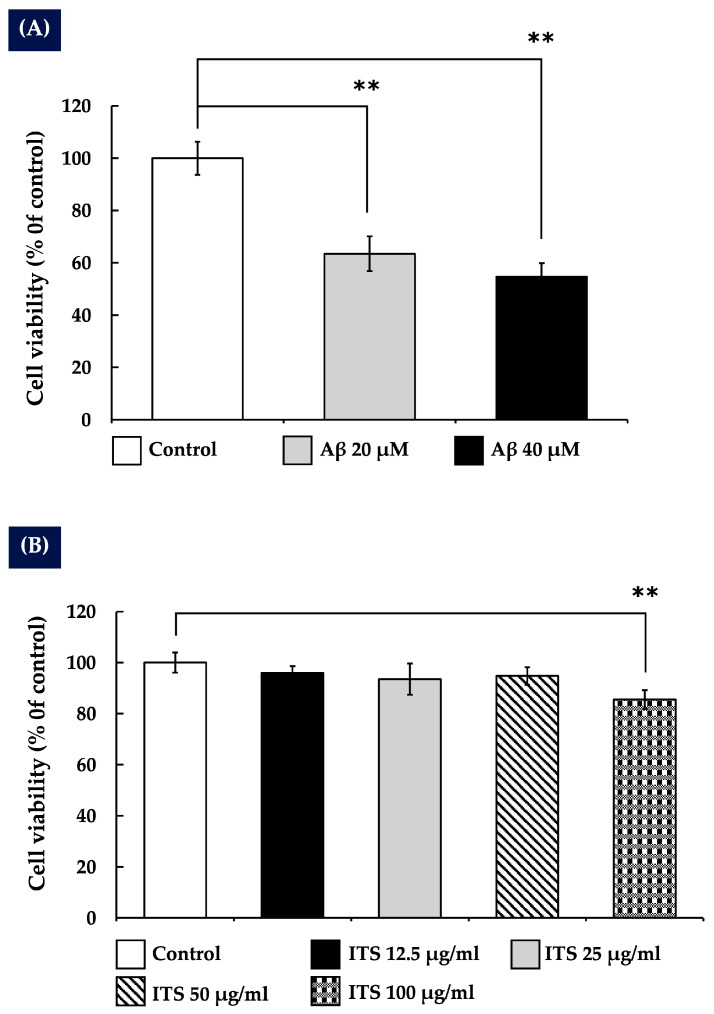
(**A**) The effect of Aβ_(25–35)_ on the viability of SH-SY5Y cells. Cells were treated with Aβ_(25–35)_ for 24 h, and cell viability was assessed using the MTT assay. (**B**) The effect of ITS on SH-SY5Y cell viability. Cells were treated with ITS for 24 h, and viability was measured using the MTT assay. Data are presented as the mean ± SEM from three independent experiments. ** *p* < 0.01 compared with the control group. Aβ, amyloid beta; ITS, seed extract of the Indian trumpet tree; MTT, (3-(4,5-dimethylthiazol-2-yl)-2,5-diphenyltetrazolium bromide).

**Figure 2 ijms-26-06288-f002:**
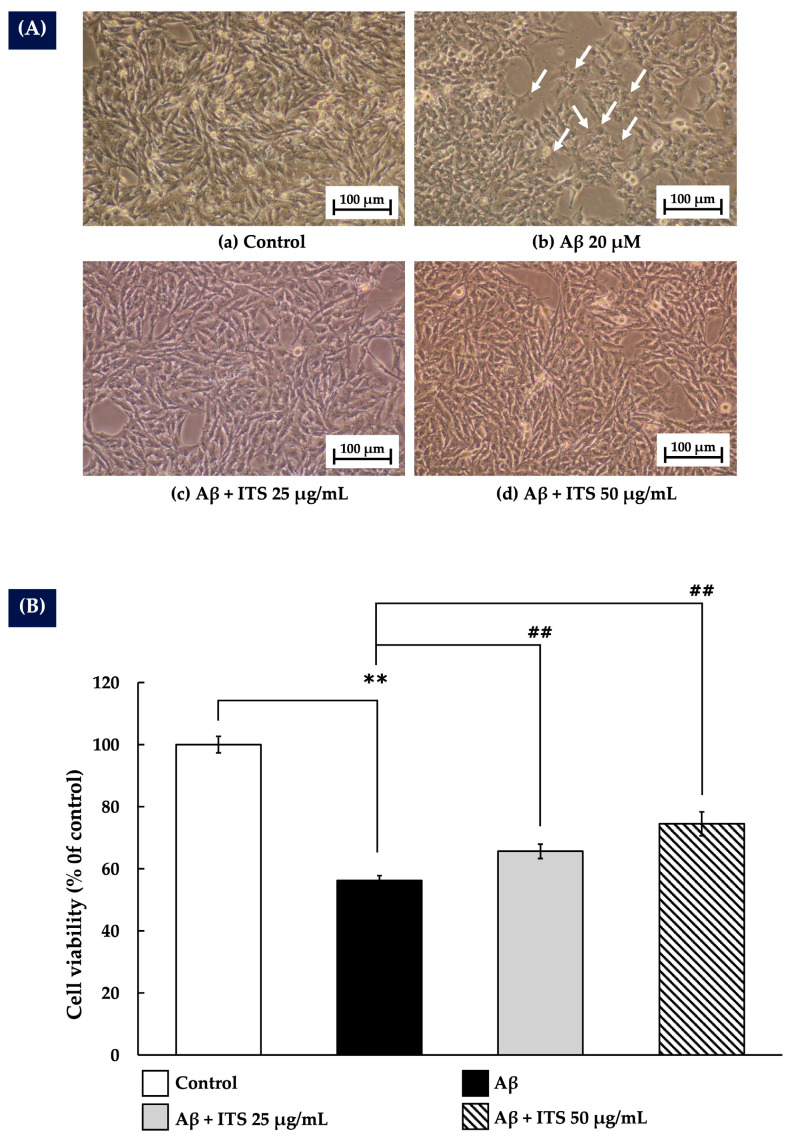
Protective effect of ITS against Aβ-induced cytotoxicity in SH-SY5Y cells after 24 h of treatment. (**A**) Light microscopy images showing SH-SY5Y cell morphology at 10× magnification. Arrows indicate neuronal damage. (**B**) Percentage of cell viability assessed using the MTT assay. Data are presented as the mean ± SEM from three independent experiments. ** *p* < 0.01 compared with the control group; ^##^ *p* < 0.01 compared with the Aβ-treated group. Aβ, amyloid beta; ITS, seed extract of Indian trumpet tree.

**Figure 3 ijms-26-06288-f003:**
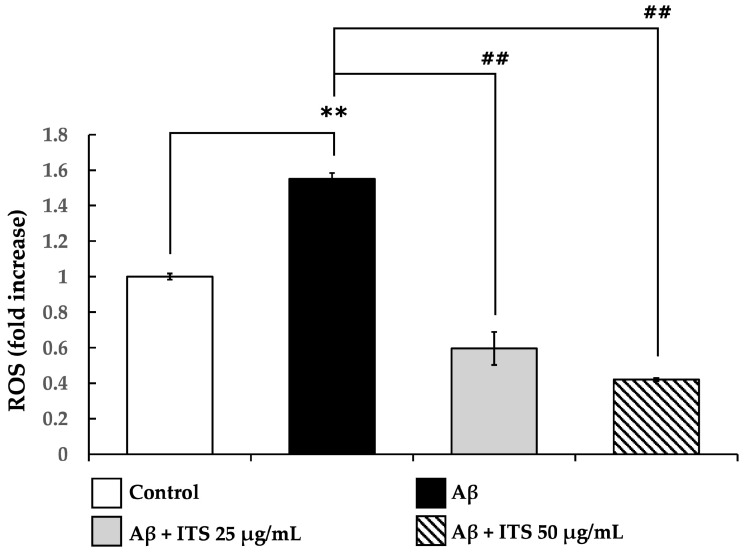
The effect of ITS on Aβ-induced ROS production in SH-SY5Y cells, measured using the fluorescent probe 2′,7′-dichlorofluorescein. Data are presented as the mean ± SEM from three independent experiments. ** *p* < 0.01 compared with the control group; ^##^ *p* < 0.01 compared with the Aβ-treated group. Aβ, amyloid beta; ITS, seed extract of the Indian trumpet tree.

**Figure 4 ijms-26-06288-f004:**
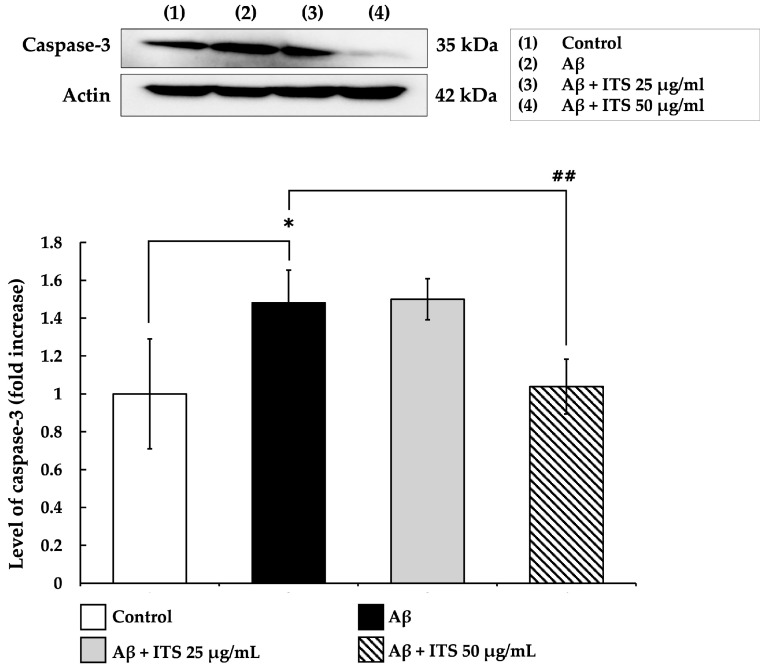
Effect of ITS on Aβ-induced caspase-3 expression in SH-SY5Y cells. Data are presented as the mean ± SEM from three independent experiments. * *p* < 0.05 compared with the control group; ^##^ *p* < 0.01 compared with the Aβ-treated group. Aβ, amyloid beta; ITS, seed extract of Indian trumpet tree.

**Figure 5 ijms-26-06288-f005:**
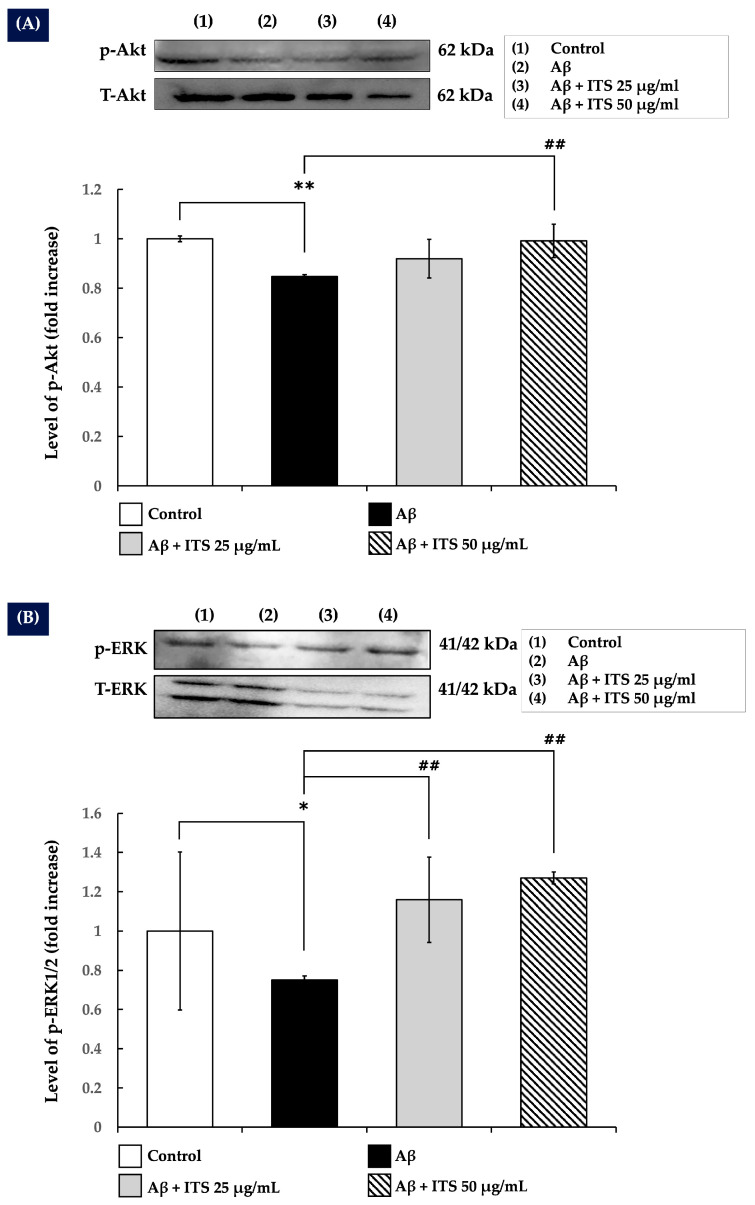
The effect of ITS on the levels of (**A**) p-Akt and (**B**) p-ERK1/2 in SH-SY5Y cells. Following treatment, total cell lysates were collected, and the levels of p-Akt and p-ERK1/2 were analyzed by Western blotting. The histograms show fold changes in protein expression relative to the untreated control group. Data are presented as the mean ± SEM of three independent experiments. * *p* < 0.05, ** *p* < 0.01 compared with the control group; ^##^ *p* < 0.01 compared with the Aβ-treated group. Aβ, amyloid beta; ITS, seed extract of the Indian trumpet tree; p-, phosphorylated.

**Figure 6 ijms-26-06288-f006:**
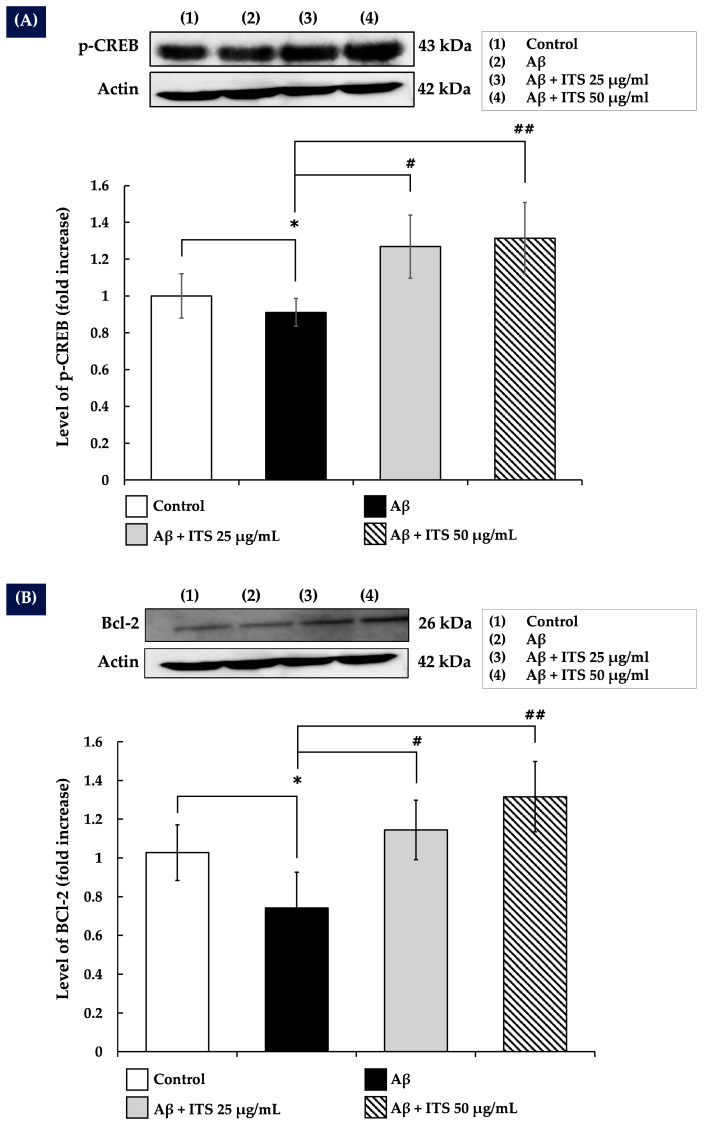
The effect of ITS on (**A**) p-CREB and (**B**) Bcl-2 levels in SH-SY5Y cells. Following treatment, total cell lysates were collected, and the levels of p-CREB and Bcl-2 were analyzed by Western blotting. The histograms show fold changes in protein expression relative to the untreated control group. Data are presented as the mean ± SEM of three independent experiments. * *p* < 0.05 compared with the control group; ^#^ *p* < 0.05, ^##^ *p* < 0.01 compared with the Aβ-treated group. Aβ, amyloid beta; CREB, cAMP-responsive element-binding protein; ITS, seed extract of the Indian trumpet tree; p-, phosphorylated.

**Figure 7 ijms-26-06288-f007:**
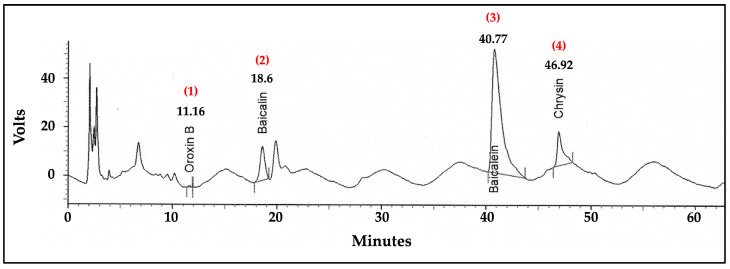
HPLC of ITS extract. Peak 1 = oroxin B; peak 2 = baicalin; peak 3 = baicalein; peak 4 = chrysin. ITS, seed extract of the Indian trumpet tree.

**Figure 8 ijms-26-06288-f008:**
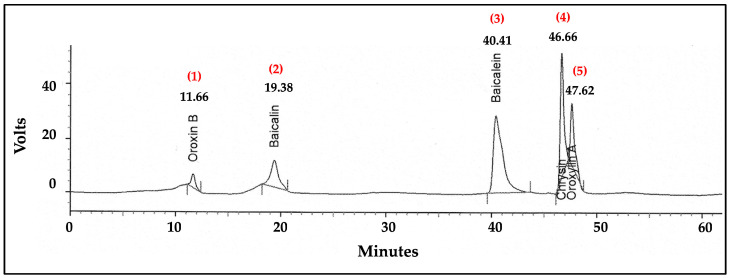
HPLC of standard compounds: (1) oroxin B, (2) baicalin, (3) baicalein, (4) chrysin, and (5) oroxylin A.

**Figure 9 ijms-26-06288-f009:**
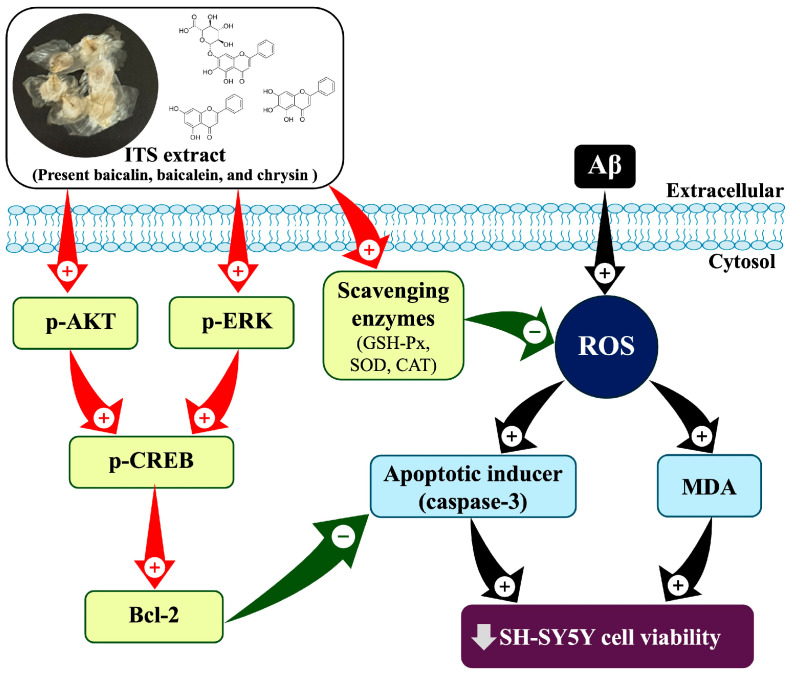
Possible mechanism of ITS-mediated neuroprotective effects against Aβ-induced cell injury. Aβ, amyloid beta; ITS, seed extract of Indian trumpet tree; p-, phosphorylated.

**Table 1 ijms-26-06288-t001:** Effects of ITS on oxidative stress markers in SH-SY5Y cells exposed to Aβ_(25–35)_-induced cytotoxicity.

Treatment Groups	MDA (nmol/mg Protein)	CAT Activity (U/mg Protein)	SOD Activity (%Inhibition/ mg Protein)	GSH-Px Activity (U/mg Protein)
Control	0.771	23.2816	43.248	0.3863
Aβ	0.97 *	24.0079	30.893 *	0.2022 *
Aβ + ITS 25 m/mL	0.755	32.1984 ^#^	41.372	0.3087 ^#^
Aβ + ITS 50 m/mL	0.514 ^#^	58.05310 ^##^	54.304 ^##^	0.4008 ^##^

The data are presented as the mean ± SEM from three independent experiments. * *p* < 0.05 compared with the control, ^#^ *p* < 0.05 and ^##^ *p* < 0.01 compared with SH-SY5Y cells treated with Aβ. Aβ, amyloid beta; ITS, seed extract of the Indian trumpet tree; MDA, malondialdehyde; CAT, catalase; SOD, superoxide dismutase; GSH-Px, glutathione peroxidase; U, units.

## Data Availability

The original contributions presented in this study are included in the article and Appendix A. Further inquiries can be directed at the corresponding authors.

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
