# Peer review of "The Neuroprotective Potential of Seed Extract from the Indian Trumpet Tree Against Amyloid Beta-Induced Toxicity in SH-SY5Y Cells"

_ijms, 2025, doi:10.3390/ijms26136288_

Round 1
Reviewer 1 Report
Comments and Suggestions for Authors
In my opinion, the manuscript can be accepted for publication after solving the main issues:
1) The introduction would benefit from improved logical flow and conceptual clarity. The transition from the general overview of Alzheimer's disease to the discussion of Oroxylum indicum is abrupt. In my opinion the authors should clearly establish the rationale for studying flavonoids in the context of AD, linking more explicitly oxidative stress, Aβ-induced neurotoxicity and the reported neuroprotective properties of baicalin and baicalein. Furthermore, the aim of the study should be stated more clearly and earlier in the introduction, instead of only being introduced in the final paragraph.
2) The introduction lacks a clear justification for the choice of baicalin and baicalein as candidate compounds for Alzheimer's disease. Although their anti-inflammatory and antidepressant effects are mentioned, the connection to AD-specific molecular mechanisms, such as Aβ-induced oxidative stress, mitochondrial dysfunction or tau pathology, is only superficially addressed. A stronger rationale is needed to support the relevance of these flavonoids in the SH-SY5Y/Aβ model, ideally by referring to previous evidence of their neuroprotective effects in AD-relevant systems.
3) The authors use fragment Aβ 25-35 as an in vitro model. However, the authors do not provide a rationale for this choice. It would be important to clarify why this fragment was chosen rather than Aβ1-42, which is more commonly associated with the pathophysiology of AD and is widely used in experimental models.
4) In general, the discussion section would benefit from more concise writing. Many periods repeat basic information that is already in the introduction or could be moved to the introduction. More space for innovation of results.
5) Considering that the aggregation state of amyloid-beta (Aβ) critically influences its neurotoxicity, the authors considered assessing whether ITS extract affects the aggregation or oligomerization of Aβ₍₂₅-₃₅₎? Evaluation of this aspect could provide valuable mechanistic insights into whether the observed neuroprotective effects are due to direct modulation of Aβ aggregation or to downstream cellular effects.

Author Response
Response to reviewer and editor suggestion
We sincerely appreciate your letter and the reviewers’ thoughtful and constructive comments on our manuscript, “Neuroprotective Potential of Seed Extract from the Indian Trumpet Tree Against β-Amyloid-Induced Toxicity in SH-SY5Y Cells” (Manuscript ID: ijms-3709159).
We are grateful for the opportunity to revise our manuscript. The reviewers’ feedback has been invaluable in enhancing the scientific quality and clarity of our work. We apologize for any oversights in the original submission.
We have carefully addressed all the comments and revised the manuscript accordingly. Below is a detailed of the major revisions made in response to the reviewers’ suggestions.
Additionally, as the reviewers noted that the manuscript required improvements in English language and style, we have used the English editing services provided by MDPI to enhance the clarity and readability of the text.
Reviewer 1
In my opinion, the manuscript can be accepted for publication after solving the main issues:
Response: We sincerely thank the reviewer for the positive evaluation and constructive comments. We have carefully addressed all the main issues raised, and revised the manuscript accordingly. We believe these revisions have significantly improved the quality and clarity of the work, and we hope the revised version meets the requirements for publication.
Comment 1: The introduction would benefit from improved logical flow and conceptual clarity. The transition from the general overview of Alzheimer's disease to the discussion of Oroxylum indicum is abrupt. In my opinion the authors should clearly establish the rationale for studying flavonoids in the context of AD, linking more explicitly oxidative stress, Aβ-induced neurotoxicity and the reported neuroprotective properties of baicalin and baicalein. Furthermore, the aim of the study should be stated more clearly and earlier in the introduction, instead of only being introduced in the final paragraph.
Response 1: We sincerely thank the reviewer for the insightful and constructive suggestions regarding the structure and clarity of the introduction. In the revised manuscript, we have improved the logical flow by restructuring the section to provide a smoother. Additionally, we have revised the introduction to clearly state the aim of the study earlier in the section, as recommended (as showed in introduction part).
Comment 2: The introduction lacks a clear justification for the choice of baicalin and baicalein as candidate compounds for Alzheimer's disease. Although their anti-inflammatory and antidepressant effects are mentioned, the connection to AD-specific molecular mechanisms, such as Aβ-induced oxidative stress, mitochondrial dysfunction or tau pathology, is only superficially addressed. A stronger rationale is needed to support the relevance of these flavonoids in the SH-SY5Y/Aβ model, ideally by referring to previous evidence of their neuroprotective effects in AD-relevant systems.
Response 2: We sincerely thank the reviewer for this valuable suggestion. We have revised the introduction to incorporate these points and improve the overall clarity and rationale of the study. (as showed in introduction part).
Comment 3: The authors use fragment Aβ 25-35 as an in vitro model. However, the authors do not provide a rationale for this choice. It would be important to clarify why this fragment was chosen rather than Aβ1-42, which is more commonly associated with the pathophysiology of AD and is widely used in experimental models.
Response 3: We thank the reviewer for this valuable observation. We agree that providing a rationale for the use of Aβ25–35 is important. In the revised manuscript, we have added a justification for this choice (in the last paragraph of introduction).
Comment 4: In general, the discussion section would benefit from more concise writing. Many periods repeat basic information that is already in the introduction or could be moved to the introduction. More space for innovation of results.
Response 4: We thank the reviewer for the thoughtful feedback regarding the discussion section. In the revised manuscript, we have carefully edited the discussion to improve conciseness by removing repetitive content and removed some sentences to the introduction section. (red fronts in discussion).
Comment 5: Considering that the aggregation state of amyloid-beta (Aβ) critically influences its neurotoxicity, the authors considered assessing whether ITS extract affects the aggregation or oligomerization of Aβ₍₂₅-₃₅₎? Evaluation of this aspect could provide valuable mechanistic insights into whether the observed neuroprotective effects are due to direct modulation of Aβ aggregation or to downstream cellular effects.
Response 5: We thank the reviewer for this insightful comment. We agree that the aggregation state of Aβ plays a critical role in its neurotoxicity, and evaluating whether ITS extract affects Aβ₍₂₅–₃₅₎ aggregation or oligomerization would indeed provide valuable mechanistic insights. We acknowledge this as an important direction for future research, and we plan to investigate the potential of ITS extract to modulate Aβ aggregation in subsequent studies.

Reviewer 2 Report
Comments and Suggestions for Authors
The manuscript entitled “Neuroprotective Potential of Seed Extract from the Indian Trumpet Tree Against β-Amyloid-Induced Toxicity in SH-SY5Y Cells” presents a compelling investigation into the neuroprotective effects of Indian trumpet tree seed extract (ITS) against β-amyloid (Aβ)-induced toxicity in SH-SY5Y cells, offering insights into its potential for Alzheimer’s disease (AD) therapy. Key strengths include a robust experimental design, clear presentation of results, and identification of bioactive flavonoids via HPLC analysis. However, the manuscript requires major revisions to strengthen the scientific context.
- Title: The current title is informative but could be made more concise and specific to enhance readability and impact. Recommended Title: “Neuroprotective Effects of Oroxylum indicum Seed Extract Against β-Amyloid Toxicity in SH-SY5Y Cells”
- Abstract: The abstract omits the specific concentrations of the Oroxylum indicum seed extract (ITS) used (25 and 50 µg/mL). Including these concentrations would enhance transparency and allow readers to better understand the experimental framework. The conclusion currently implies translational potential, which may overstate the findings given the in vitro nature of the study. It would be more appropriate to temper this statement by acknowledging the need for in vivo validation to support clinical relevance.
- Keywords: The selected keywords are relevant and accurately reflect the core themes of the study. However, alignment with Medical Subject Headings (MeSH) terminology would enhance the manuscript’s visibility and searchability in biomedical databases.
- Introduction: Authors may consider restructuring the introduction to incorporate: (1) recent AD prevalence data to underscore its global significance; (2) an overview of AD models, with emphasis on the SH-SY5Y cell model; (3) current therapeutic challenges; (4) the role of herbal medicine, first highlighting Oroxylum indicum seed extract and then its main ingredients, such as baicalin’s neuroprotective properties; and (5) the relevant molecular pathways (Akt, ERK1/2, CREB, Bcl-2) involved in AD treatment. In the final paragraph, the authors could clearly articulate the study’s aim, novelty, and methodology as standalone sentences, without citing any references, to better frame the research approach. (Authors can use the list of references to update and restructure their introduction.)
- Methods: The methods are detailed and generally reproducible, covering cell culture, extraction, assays, and statistical analysis. However, the rationale for selecting ITS concentrations (25 and 50 µg/mL) is not explained, and the sample size for experiments (e.g., number of replicates) is unclear. In addition, the reason for using the SH-SY5Y cell line should be provided. Section 4.8. Analysis of flavonoid contents in crude extracts by HPLC should be placed after Chemical Plant Material and Extraction. Before this section, a brief overview of the study should be included, along with the ethical approvals and protocols, specifying which committee or institution granted approval.
- Results: The results are logically presented, with clear figures and tables that support the narrative. However, the order of the results should be rearranged to correspond with the sequence outlined in the Methods section. The HPLC findings are robust, but the discussion omits mention of minor peaks or other potentially bioactive compounds, which could be relevant. All p-values should be clearly defined and reported for each result to ensure statistical transparency. Additionally, in the MTT assay figures, representative images of the cells should be annotated with arrows to distinguish between healthy and dead cells, enhancing visual clarity and interpretation.
- Discussion: The discussion begins with a clear summary of the study’s aim and key findings but remains underdeveloped in its comparison with existing literature. For instance, comparisons with other studies on flavonoid-based neuroprotection are limited. While the mechanistic interpretations, such as the involvement of the Akt and CREB pathways, are sound, they lack depth regarding how specific ITS flavonoids contribute to these effects. Additionally, some overgeneralized claims about the therapeutic potential are made without acknowledging translational challenges. To strengthen the discussion, the authors should expand comparisons with recent flavonoid-related studies, elaborate on the specific roles of key flavonoids such as baicalin, and moderate therapeutic claims by emphasizing the need for further in vivo validation and clinical translation. (Authors can use the list of references to update and restructure their discussion.)
- Conclusion: The conclusion effectively restates the study’s aim and major findings; however, it does not sufficiently highlight the clinical relevance or outline future research directions.
- Limitations: The manuscript does not explicitly address its limitations, which is a significant oversight. Several important issues should be acknowledged, including the limited translational relevance of the in vitro SH-SY5Y cell model, the absence of in vivo validation, and potential variability in the composition of the ITS extract. Additionally, the study does not isolate or directly evaluate the specific effects of baicalin, one of the major active flavonoids.
- References: The references provided in the manuscript are relevant and consistently formatted, supporting the study’s focus on AD, Aβ\induced neurotoxicity, and the neuroprotective potential of Indian trumpet tree seed extract (ITS). However, the Introduction and Discussion sections would benefit from incorporating additional recent and high-impact studies to strengthen the scientific foundation, particularly to enhance comparisons with other neuroprotective or flavonoid-based studies. Below is a refined list of five recommended studies from the provided list to bolster the manuscript’s literature support.
Luo, H., et al. (2019). Apelin-13 suppresses neuroinflammation against cognitive deficit in a streptozotocin-induced rat model of Alzheimer’s disease through activation of BDNF-TrkB signaling pathway. Frontiers in Pharmacology, 10, 395. doi: 10.3389/fphar.2019.00395
Li, H., et al. (2022). Untargeted metabolomics analysis of the hippocampus and cerebral cortex identified the neuroprotective mechanisms of Bushen Tiansui formula in an Aβ25-35-induced rat model of Alzheimer's disease. Frontiers in Pharmacology, 13, 990307. doi: 10.3389/fphar.2022.990307
Xiang, Q., et al. (2024). Revealing the potential therapeutic mechanism of Lonicerae Japonicae Flos in Alzheimer’s disease: a computational biology approach. Frontiers in Medicine, 11, 1468561. doi: 10.3389/fmed.2024.1468561
Cheng, X., et al. (2024). Quercetin: A promising therapy for diabetic encephalopathy through inhibition of hippocampal ferroptosis. Phytomedicine, 126, 154887. doi: 10.1016/j.phymed.2023.154887
Hui, Z., et al. (2024). Mechanisms and therapeutic potential of chinonin in nervous system diseases. Journal of Asian Natural Products Research, 26(12), 1405–1420. doi: 10.1080/10286020.2024.2371040
Kang, L., et al. (2018). Structure–activity relationship investigation of coumarin–chalcone hybrids with diverse side-chains as acetylcholinesterase and butyrylcholinesterase inhibitors. Molecular Diversity, 22(4), 893–906. doi: 10.1007/s11030-018-9839-y
Gao, X., et al. (2019). Structure–activity study of fluorine or chlorine-substituted cinnamic acid derivatives with tertiary amine side chain in acetylcholinesterase and butyrylcholinesterase inhibition. Drug Development Research, 80(4), 438–445. doi: 10.1002/ddr.21515
Lu, Q., et al. (2020). Nitrogen-containing flavonoid and their analogs with diverse B-ring in acetylcholinesterase and butyrylcholinesterase inhibition. Drug Development Research, 81(8), 1037–1047. doi: 10.1002/ddr.21726
- Tables: Table 1 is well-formatted and clearly presents oxidative stress marker data, complementing the narrative. However, the table could include a footnote explaining the units (e.g., U/mg protein for CAT) for non-specialist readers.
Additional General Considerations: The manuscript contains minor typographical errors and inconsistent use of abbreviations. While grammatical clarity is generally acceptable throughout the manuscript, the Discussion section would benefit from further polishing to improve sentence flow and coherence. All abbreviations should be defined upon first use in both the abstract and the main text.
Comments on the Quality of English LanguageAdditional General Considerations: The manuscript contains minor typographical errors and inconsistent use of abbreviations. While grammatical clarity is generally acceptable throughout the manuscript, the Discussion section would benefit from further polishing to improve sentence flow and coherence. All abbreviations should be defined upon first use in both the abstract and the main text.
Author Response
Response to reviewer and editor suggestion
We sincerely appreciate your letter and the reviewers’ thoughtful and constructive comments on our manuscript, “Neuroprotective Potential of Seed Extract from the Indian Trumpet Tree Against β-Amyloid-Induced Toxicity in SH-SY5Y Cells” (Manuscript ID: ijms-3709159).
We are grateful for the opportunity to revise our manuscript. The reviewers’ feedback has been invaluable in enhancing the scientific quality and clarity of our work. We apologize for any oversights in the original submission.
We have carefully addressed all the comments and revised the manuscript accordingly. Below is a detailed of the major revisions made in response to the reviewers’ suggestions.
Additionally, as the reviewers noted that the manuscript required improvements in English language and style, we have used the English editing services provided by MDPI to enhance the clarity and readability of the text.
Reviewer 2
The manuscript entitled “Neuroprotective Potential of Seed Extract from the Indian Trumpet Tree Against β-Amyloid-Induced Toxicity in SH-SY5Y Cells” presents a compelling investigation into the neuroprotective effects of Indian trumpet tree seed extract (ITS) against β-amyloid (Aβ)-induced toxicity in SH-SY5Y cells, offering insights into its potential for Alzheimer’s disease (AD) therapy. Key strengths include a robust experimental design, clear presentation of results, and identification of bioactive flavonoids via HPLC analysis. However, the manuscript requires major revisions to strengthen the scientific context.
Response: We sincerely appreciate your valuable feedback and your recognition of the strengths of our manuscript. In response to your recommendation for major revisions to enhance the scientific context, we have thoroughly addressed all key concerns and revised the manuscript accordingly. We believe that these revisions have substantially improved the clarity, coherence, and overall quality of the work, and we hope that the updated version will meet the standards for publication.
Comment 1: Title: The current title is informative but could be made more concise and specific to enhance readability and impact. Recommended Title: “Neuroprotective Effects of Oroxylum indicum Seed Extract Against β-Amyloid Toxicity in SH-SY5Y Cells”
Response 1: Thank you for your thoughtful comment regarding the manuscript title. We truly appreciate your suggestion to revise it for greater conciseness and specificity. However, we kindly request to retain our original title, “The Neuroprotective Potential of Seed Extract from the Indian Trumpet Tree Against β-Amyloid-Induced Toxicity in SH-SY5Y Cells,” for the following reasons:
- Avoidance of self-plagiarism: In our previous publications, we have extensively used the scientific name Oroxylum indicum in titles and main texts, particularly in studies investigating the plant’s neuroprotective effects against β-amyloid-induced toxicity in SH-SY5Y cells. Although those studies focused on different parts of the plant—such as the leaf extract (Palachai et al., 2025 and Mairuae et al., 2019)—the overall research context and terminology remain closely related. As a result, reusing the scientific name in the current manuscript title contributes to a high text similarity score in plagiarism detection software (e.g., Turnitin), which may lead to concerns about self-plagiarism or reduced manuscript originality. To address this, we have used the common English name “Indian Trumpet Tree” to refer to the same plant species. This approach significantly reduces repetition while preserving clarity and scientific accuracy.
References:
- Palachai N, Buranrat B, Noisa P, Mairuae N. Oroxylum indicum (L.) leaf extract attenuates β-amyloid-induced neurotoxicity in SH-SY5Y cells. Int J Mol Sci. 2025;26(7):2917. https://doi.org/10.3390/ijms26072917
- Mairuae N, Connor JR, Buranrat B, Lee SY. Oroxylum indicum (L.) extract protects human neuroblastoma SH‑SY5Y cells against β‑amyloid‑induced cell injury. Mol Med Rep. 2019;20(2):1933–1942. https://doi.org/10.3892/mmr.2019.10411
- Scientific clarity and reader accessibility: While Oroxylum indicum is the accepted scientific name, the use of “Indian Trumpet Tree” also conveys the same meaning and is recognizable to a broader interdisciplinary readership, including those from related but non-botanical fields.
- Consistency with broader communication standards: Using the common name aligns with current trends in science communication, particularly in translational and functional food research, where the vernacular name can enhance reader engagement without compromising scientific accuracy.
We hope this explanation clarifies our choice and that the original title remains acceptable for publication. Nevertheless, we are open to making minor adjustments if required for consistency with the journal’s editorial style.
Comment 2: Abstract: The abstract omits the specific concentrations of the Oroxylum indicum seed extract (ITS) used (25 and 50 µg/mL). Including these concentrations would enhance transparency and allow readers to better understand the experimental framework. The conclusion currently implies translational potential, which may overstate the findings given the in vitro nature of the study. It would be more appropriate to temper this statement by acknowledging the need for in vivo validation to support clinical relevance.
Response 2: Thank you so much for your kind feedback. We agree with your proposal to omit the particular concentrations of ITS employed in our investigation. To promote clarity and transparency, we have included the concentrations (25 and 50 µg/mL) in the abstract (as red fronts).
We also recognize your issue about the wording of the abstract's conclusion, which may have implied broader translational implications than are proper for an in vitro investigation. We amended the conclusion to better reflect the scope of our findings, highlighting the early nature of the data and the need for additional in vivo investigations to validate clinical relevance (as highlighted in red).
Comment 3: Keywords: The selected keywords are relevant and accurately reflect the core themes of the study. However, alignment with Medical Subject Headings (MeSH) terminology would enhance the manuscript’s visibility and searchability in biomedical databases.
Response 3: Thank you very much for your valuable suggestion regarding the keywords. We have carefully reviewed and revised the keywords to better align with Medical Subject Headings (MeSH) terminology to enhance the manuscript’s visibility and searchability in biomedical databases.
Keywords:
Alzheimer’s disease;
Indian trumpet tree;
Oroxylum indicum (L.) seed;
Neuroprotection;
Oxidative stress;
Plant extraction;
Protein Kinase B;
cAMP Response Element-Binding Protein;
Mitogen-Activated Protein Kinases
Comment 4: Introduction: Authors may consider restructuring the introduction to incorporate: (1) recent AD prevalence data to underscore its global significance; (2) an overview of AD models, with emphasis on the SH-SY5Y cell model; (3) current therapeutic challenges; (4) the role of herbal medicine, first highlighting Oroxylum indicum seed extract and then its main ingredients, such as baicalin’s neuroprotective properties; and (5) the relevant molecular pathways (Akt, ERK1/2, CREB, Bcl-2) involved in AD treatment. In the final paragraph, the authors could clearly articulate the study’s aim, novelty, and methodology as standalone sentences, without citing any references, to better frame the research approach. (Authors can use the list of references to update and restructure their introduction.)
Response 4: We express our sincere gratitude to the reviewer for their insightful comments on the introduction's structure. Their valuable guidance has significantly enhanced the clarity and organization of our manuscript. We have carefully incorporated the reviewer's suggestions into the revised introduction. However, information related to Akt, ERK1/2, CREB, and Bcl-2 has been included in the Discussion section.
Comment 5: Methods: The methods are detailed and generally reproducible, covering cell culture, extraction, assays, and statistical analysis. However, the rationale for selecting ITS concentrations (25 and 50 µg/mL) is not explained, and the sample size for experiments (e.g., number of replicates) is unclear. In addition, the reason for using the SH-SY5Y cell line should be provided. Section 4.8. Analysis of flavonoid contents in crude extracts by HPLC should be placed after Chemical Plant Material and Extraction. Before this section, a brief overview of the study should be included, along with the ethical approvals and protocols, specifying which committee or institution granted approval.
Response 5: Thank you for your valuable feedback regarding our manuscript. We appreciate your comments on the methods section and have addressed the concerns raised.
Regarding the rationale for selecting ITS concentrations of 25 and 50 µg/mL, we initially conducted a toxicity screening using a range of concentrations from 0 to 100 µg/mL, as outlined in the methods. The specific concentrations of 25 and 50 µg/mL were chosen based on these results to ensure non-toxicity to the SH-SY5Y cell line, which is detailed in the results section (as red fronts).
The sample size is indicated in the figure legends within the Results section for clarity, as: 'Data are presented as the mean ± SEM of three independent experiments (as red fronts).
In the revised manuscript, the rationale for using the SH-SY5Y cell line has been provided in the last paragraph of the Introduction section.
Since the Results section has been structured to follow the sequence of the Methods (as the reviewer suggests), we respectfully propose retaining the current placement of the HPLC analysis to preserve consistency and coherence between the methodology and the corresponding results.
An overview of the study, including its aim, has been provided in the last paragraph of the Introduction section.
Thank you for your comment regarding ethical approvals and protocols. As this study did not involve animal or human subjects, ethical approval was not required. Therefore, this information was not included in the relevant section of the manuscript.
Comment 6: Results: The results are logically presented, with clear figures and tables that support the narrative. However, the order of the results should be rearranged to correspond with the sequence outlined in the Methods section. The HPLC findings are robust, but the discussion omits mention of minor peaks or other potentially bioactive compounds, which could be relevant. All p-values should be clearly defined and reported for each result to ensure statistical transparency. Additionally, in the MTT assay figures, representative images of the cells should be annotated with arrows to distinguish between healthy and dead cells, enhancing visual clarity and interpretation.
Response 6: Thank you very much for your thoughtful and constructive comments regarding the presentation of the results.
We appreciate your suggestion to align the order of the Results section with that of the Methods section. In this revised manuscript, the Results have been arranged accordingly to match the sequence presented in the Methods.
Regarding the HPLC analysis, we acknowledge the presence of several minor peaks in the chromatogram of the ITS extract. These may represent other potentially bioactive compounds, and we agree that this warrants further investigation. We have now included a sentence in the Discussion section to address this point (as red fronts).
In response to your comment on statistical reporting, we have carefully reviewed and revised the manuscript to ensure that all p-values are clearly defined and reported for each result to enhance statistical transparency.
Lastly, for the MTT assay figures, we have added representative images of the cells with annotated arrows to distinguish between healthy and dead cells.
Thank you again for your insightful feedback, which has helped us improve the clarity and quality of our manuscript.
Comment 7: Discussion: The discussion begins with a clear summary of the study’s aim and key findings but remains underdeveloped in its comparison with existing literature. For instance, comparisons with other studies on flavonoid-based neuroprotection are limited. While the mechanistic interpretations, such as the involvement of the Akt and CREB pathways, are sound, they lack depth regarding how specific ITS flavonoids contribute to these effects. Additionally, some overgeneralized claims about the therapeutic potential are made without acknowledging translational challenges. To strengthen the discussion, the authors should expand comparisons with recent flavonoid-related studies, elaborate on the specific roles of key flavonoids such as baicalin, and moderate therapeutic claims by emphasizing the need for further in vivo validation and clinical translation. (Authors can use the list of references to update and restructure their discussion.)
Response 7: We truly appreciate the reviewer's thoughtful and constructive input on the Discussion area. We revised the discussion to provide a more extensive comparison with existing literature, including studies on flavonoid-based neuroprotection. In addition, we have adjusted our remarks on the extract's medicinal potential to minimize overgeneralization. The revised text now contains a more balanced viewpoint that emphasizes the limitations of existing in vitro findings and the need for more in vivo studies. These revisions are intended to increase the scientific rigor and contextual depth of the debate, as indicated.
Comment 8: Conclusion: The conclusion effectively restates the study’s aim and major findings; however, it does not sufficiently highlight the clinical relevance or outline future research directions.
Response 8: We thank the reviewer for this insightful comment. We have revised the Conclusion section to better highlight the potential clinical relevance of our findings and to clearly outline future research directions. Specifically, we now emphasize the need for in vivo validation, and the isolation of active compounds such as baicalin and baicalein to further explore their individual contributions to the observed neuroprotective effects. As red fronts in conclusion.
“Future research should examine the effects of ITS on in vivo models of AD in order to further evaluate its therapeutic significance. Furthermore, separating active ingredients like baicalin and baicalein for individual analysis may assist elucidate their distinct roles. These efforts would support the development of ITS-based interventions as potential candidates for AD treatment.”
Comment 9: Limitations: The manuscript does not explicitly address its limitations, which is a significant oversight. Several important issues should be acknowledged, including the limited translational relevance of the in vitro SH-SY5Y cell model, the absence of in vivo validation, and potential variability in the composition of the ITS extract. Additionally, the study does not isolate or directly evaluate the specific effects of baicalin, one of the major active flavonoids.
Response 9: We sincerely thank the reviewer for highlighting the important point regarding the limitations of our study. We agree that acknowledging these limitations strengthens the manuscript. Accordingly, we have added a dedicated section in the revised manuscript to discuss them (as red fronts in the last paragraph of discussion).
“The limitation of this study is that it was conducted in vitro using the SH-SY5Y cell model. While this model is widely used for studying neurodegenerative mechanisms, it does not fully replicate the complexity of neuronal networks or the in vivo environment. Therefore, further studies using animal models are essential to validate the neuroprotective potential of ITS extract in a more biologically relevant context. Moreover, future research should focus on isolating baicalin and baicalein from ITS extract to specifically investigate their individual neuroprotective effects.”
Comment 10: References: The references provided in the manuscript are relevant and consistently formatted, supporting the study’s focus on AD, Aβ\induced neurotoxicity, and the neuroprotective potential of Indian trumpet tree seed extract (ITS). However, the Introduction and Discussion sections would benefit from incorporating additional recent and high-impact studies to strengthen the scientific foundation, particularly to enhance comparisons with other neuroprotective or flavonoid-based studies. Below is a refined list of five recommended studies from the provided list to bolster the manuscript’s literature support.
Luo, H., et al. (2019). Apelin-13 suppresses neuroinflammation against cognitive deficit in a streptozotocin-induced rat model of Alzheimer’s disease through activation of BDNF-TrkB signaling pathway. Frontiers in Pharmacology, 10, 395. doi: 10.3389/fphar.2019.00395
Li, H., et al. (2022). Untargeted metabolomics analysis of the hippocampus and cerebral cortex identified the neuroprotective mechanisms of Bushen Tiansui formula in an Aβ25-35-induced rat model of Alzheimer's disease. Frontiers in Pharmacology, 13, 990307. doi: 10.3389/fphar.2022.990307
Xiang, Q., et al. (2024). Revealing the potential therapeutic mechanism of Lonicerae Japonicae Flos in Alzheimer’s disease: a computational biology approach. Frontiers in Medicine, 11, 1468561. doi: 10.3389/fmed.2024.1468561
Cheng, X., et al. (2024). Quercetin: A promising therapy for diabetic encephalopathy through inhibition of hippocampal ferroptosis. Phytomedicine, 126, 154887. doi: 10.1016/j.phymed.2023.154887
Hui, Z., et al. (2024). Mechanisms and therapeutic potential of chinonin in nervous system diseases. Journal of Asian Natural Products Research, 26(12), 1405–1420. doi: 10.1080/10286020.2024.2371040
Kang, L., et al. (2018). Structure–activity relationship investigation of coumarin–chalcone hybrids with diverse side-chains as acetylcholinesterase and butyrylcholinesterase inhibitors. Molecular Diversity, 22(4), 893–906. doi: 10.1007/s11030-018-9839-y
Gao, X., et al. (2019). Structure–activity study of fluorine or chlorine-substituted cinnamic acid derivatives with tertiary amine side chain in acetylcholinesterase and butyrylcholinesterase inhibition. Drug Development Research, 80(4), 438–445. doi: 10.1002/ddr.21515
Lu, Q., et al. (2020). Nitrogen-containing flavonoid and their analogs with diverse B-ring in acetylcholinesterase and butyrylcholinesterase inhibition. Drug Development Research, 81(8), 1037–1047. doi: 10.1002/ddr.21726
Response 10: We sincerely thank the reviewer for the constructive feedback regarding the references and literature support. We appreciate your recognition of the relevance and consistency of the current citations. We have revised both the Introduction and Discussion sections to incorporate recent and high-impact studies, particularly those related to neuroprotection and flavonoid-based interventions. While we have included three of the recommended references (as the yellow highlight above), we have also added additional up-to-date citations to further strengthen the scientific foundation. We hope these revisions enhance the manuscript’s alignment with current research and reinforce its contribution to the field.
Comment 11: Tables: Table 1 is well-formatted and clearly presents oxidative stress marker data, complementing the narrative. However, the table could include a footnote explaining the units (e.g., U/mg protein for CAT) for non-specialist readers.
Response 11: We thank the reviewer for the positive feedback on Table 1. In response to the suggestion, we have added a footnote to the table to clearly explain the units (e.g., U/mg protein for CAT and other relevant parameters), to enhance clarity for non-specialist readers.As red fronts in a footnote of table
Comment 12: Additional General Considerations: The manuscript contains minor typographical errors and inconsistent use of abbreviations. While grammatical clarity is generally acceptable throughout the manuscript, the Discussion section would benefit from further polishing to improve sentence flow and coherence. All abbreviations should be defined upon first use in both the abstract and the main text.
Response 12: We sincerely thank the reviewer for the constructive comments. We have carefully proofread the manuscript to correct minor typographical errors and ensure consistent use of abbreviations throughout. All abbreviations are now defined upon their first appearance in both the abstract and the main text. Additionally, we have revised the Discussion section to improve sentence flow and overall coherence, as suggested.

Round 2
Reviewer 1 Report
Comments and Suggestions for Authors
The authors have satisfied all my requests, so in my opinion the article can be published.
Reviewer 2 Report
Comments and Suggestions for Authors
The authors have addressed the concerns raised. However, my concern remains regarding the initial 40% duplication report. I recommend a thorough review to ensure originality before the paper can be accepted for publication.